# Heat-Killed *Bifidobacterium*
*breve* B-3 Enhances Muscle Functions: Possible Involvement of Increases in Muscle Mass and Mitochondrial Biogenesis

**DOI:** 10.3390/nu12010219

**Published:** 2020-01-15

**Authors:** Kazuya Toda, Yuki Yamauchi, Azusa Tanaka, Tetsuya Kuhara, Toshitaka Odamaki, Shin Yoshimoto, Jin-zhong Xiao

**Affiliations:** Next Generation Science Institute, Morinaga Milk Industry Co., Ltd., Kanagawa 252-8583, Japan; yuki-yamauchi045@morinagamilk.co.jp (Y.Y.); ad-tanaka@morinagamilk.co.jp (A.T.); t_kuhara@morinagamilk.co.jp (T.K.); t-odamak@morinagamilk.co.jp (T.O.); shin-yoshimoto923@morinagamilk.co.jp (S.Y.); j_xiao@morinagamilk.co.jp (J.-z.X.)

**Keywords:** skeletal muscles, probiotics, *Bifidobacterium**breve* B-3, muscle mass, mitochondria

## Abstract

A previous clinical study on pre-obesity subjects revealed that *Bifidobacterium*
*breve* B-3 shows anti-obesity effects and possibly increases muscle mass. Here, we investigated the effects of B-3 on muscle function, such as muscle strength and metabolism, and some signaling pathways in skeletal muscle. Male rodents were orally administered live B-3 (B-3L) or heat-killed B-3 (B-3HK) for 4 weeks. We found that administration of B-3 to rats tended to increase muscle mass and affect muscle metabolism, with stronger effects in the B-3HK group than in the B-3L group. B-3HK significantly increased muscle mass and activated Akt in the rat soleus. With regard to muscle metabolism, B-3HK significantly increased phosphorylated AMP-activated protein kinase (AMPK), peroxisome proliferator-activated receptor gamma coactivator (PGC)-1α and cytochrome *c* oxidase (CCO) gene expression in the rat soleus, suggesting an effect on the AMPK-PGC1α-mitochondrial biogenesis pathway. Furthermore, B-3HK promoted oxidative muscle fiber composition in the gastrocnemius. We also observed a significantly higher level of murine grip strength in the B-3HK group than in the control group. These findings suggest the potential of heat-killed B-3 in promoting muscle hypertrophy and modifying metabolic functions, possibly through the Akt and AMPK pathways, respectively.

## 1. Introduction

*Bifidobacterium breve* B-3 (B-3), a probiotic strain originating from the gut of an infant, has been demonstrated to exert anti-obesity effects [1,2] through mechanisms speculated to include improvement of intestinal barrier function; adiponectin and colonic proglucagon production; and the production of B-3-derived metabolites with anti-obesity activity (e.g., acetic acids and conjugated linoleic acids) [1,3]. In a clinical study on mild obesity subjects, body fat mass was significantly lower in the B-3 group than in the placebo group. Remarkably, B-3 administration also significantly increased muscle mass [2], suggesting potential effects of B-3 on muscle.

Accumulating evidence has indicated that gut microbiota are associated with host health conditions in numerous ways [4], including through energy metabolism and mitochondrial function [5,6]. Moreover, the cross-talk pathway between the gut microbiota and skeletal muscle, i.e., the gut-muscle axis, has been extensively studied, and microbiota composition and the intestinal environment have been suggested to influence muscle mass and function, possibly by modification of microbiota composition, immune function, energy metabolism and oxidative stress [7,8].

However, studies on the effects of probiotics on muscle mass and function have been scarcely reported, and the potential effects of probiotics on physical performance and their underlying mechanisms remain unclear. Few studies have suggested the potential impact of probiotics on the gut microbiomes of athletes [9], and the possible involvement of metabolites of gut microbiota, such as acetic acid, in the stimulation of muscular energy metabolism [10], and the enhancement of endurance performance [11].

Recently, heat-killed microorganisms have attracted attention as postbiotics [12]. Numerous studies have indicated the effects of the cell components of probiotic bacteria in modulating the immune functions and enhancing the intestinal barrier [13,14]. Piqué et al. showed that non-viable bacteria and bacterial fractions could pass through the mucus and stimulate epithelial cells more efficiently compared with viable bacteria [15]. In addition, although the use of probiotic bacteria has been demonstrated to meet safety concerns, some uses of probiotic strains have been pointed at regarding risks such as systemic infections due to translocation, particularly in vulnerable patients and pediatric populations [15]. Therefore, from a safety point of view, there is an increasing interest in non-viable beneficial microbes to be used as functional ingredients. Furthermore, heat-killed bacteria are generally easier and more suitable for industrial applications in different types of foods and dietary supplements.

We investigated whether B-3 influences muscle mass and muscle metabolism using rodents fed a regular chow diet. To understand the mechanisms, the activations of Akt and AMPK involved in the signalling pathway related to muscle mass and muscle metabolism in skeletal muscle, respectively, were evaluated [16,17]. Moreover, animals were treated with heat-killed B-3 to evaluate the potential effects of heat-killed bacteria and to understand the underlying mechanisms of the effects of B-3 on skeletal muscle.

## 2. Materials and Methods

### 2.1. Preparation of the Samples

B-3 (MCC1274) lyophilized powder was obtained from the Morinaga Milk Industry (Tokyo, Japan). The live B-3 (B-3L) were suspended in saline just before daily administration. The heat-killed B-3 (B-3HK) were prepared as previously described with slight modifications [18], by heating B-3 lyophilized powder suspended in saline at 90 °C for 30 min. A lack of viable bacteria was confirmed with anaerobic culture methods using TOS propionate agar (Eiken Chemical, Tokyo, Japan). B-3HK was stored at −20 °C until an administration.

### 2.2. Animal Experiments

All animal studies were approved by the Animal Research Committee of Morinaga Milk Industry (approval dates: 22 February 2018 and 21 September 2018) and performed in accordance with the relevant guidelines and regulations. Male, 8-week-old Crl:CD (SD) rats and C57BL/6J mice (CRJ, Inc., Kanagawa, Japan) were housed in individual cages under controlled lighting conditions (12 h light/dark cycle; lights on from 8:00 to 20:00) at a constant temperature (25 °C) and were provided Labo MR Stock food (NOSAN Corporation, Kanagawa, Japan) and water ad libitum.

In Experiment 1 (rearing date: 6 March to 19 April 2018), rats were used for evaluation of the effects of B-3 on the anabolic and catabolic signaling pathways. Fifty rats were divided into the following four groups (*n* = 12 or 13): a control group (given saline), a positive control group for mTOR activation (given leucine at 1 mmol/kg/day), a B-3L group (given 1 × 10^9^ cfu/rat) and a B-3HK group (given an amount of B-3HK equivalent to the number of cells given to the B-3L group). Each ingredient was orally administered six days a week for 28 days with the exception of the dissection date. Body weight and food intake per day were monitored weekly (Figure 1).

In Experiment 2 (rearing date: 2 October to 22 November 2018), mice were evaluated for fitness performance. Thirty-six mice were divided into 2 groups (*n* = 18): a control group (given saline) and a B-3HK group (given the same dosage as in experiment 1). The treatments were orally administered for 4 weeks, as in Experiment 1, and then a grip test was conducted at 2 and 4 weeks.

### 2.3. Western Blotting

The left solei of the rats (20 mg) were stored at −80 °C after freezing with liquid nitrogen. The frozen tissues were homogenized for 2 min on ice with RIPA buffer (Cell Signaling Technology [CST], Danvers, MA, USA), protease inhibitor (Invitrogen, Carlsbad, CA, USA) and 1 mM phenylmethylsulfonyl fluoride (PMSF, Nacalai Tesque, Inc., Kyoto, Japan) using a BioMasher (Nippi, Inc., Tokyo, Japan). After centrifugation at 13,000× *g* for 15 min at 4 °C, the protein in the supernatant was measured with a bicinchoninic acid (BCA) assay (Thermo Fisher Scientific, Waltham, MA, USA). After heat treatment for 5 min at 95 °C, the lysate samples were loaded (10 µg of protein per well) with SDS sample buffer (250 mM Tris-HCl (pH 6.8), 5% glycerol, 5% 2-mercaptoethanol, 2% SDS and 0.01% bromophenol blue) onto NuPAGE gels with MOPS buffer (Invitrogen) and separated. After SDS-PAGE, the proteins were transferred to PVDF membranes using an iBlot system (Invitrogen). The membranes were blocked for 1 h at RT with Blocking One buffer (Nacalai Tesque, Inc., Kyoto, Japan), and then incubated overnight at 4 °C with phosphorylated Akt (pAkt, catalogue number 2965, CST, Boston, MA, USA), Akt (catalogue number 2966, CST), phosphorylated mammallian target of rapamycin (pmTOR, catalogue number 5536, CST), mTOR (catalogue number 4517, CST), phosphorylated p70 S6 kinase (pp70S6K, catalogue number 9206, CST), p70S6K (catalogue number 2708, CST), phosphorylated AMP-activated protein kinase (pAMPK, catalogue number 2535, CST) and AMPK (catalogue number 2793, CST) or β-actin (catalogue number sc-47778, Santa Cruz Biotechnology, Inc., Dallas, TX, USA) antibodies at 1:1000 dilutions. The membranes were washed with TBS buffer with 1% Tween 20 and incubated for 1 h at RT with anti-rabbit or anti-mouse horseradish peroxidase-conjugated secondary antibodies (catalogue number 074-1516 or 074-1806, KPL, Gaithersburg, MD, USA) at a concentration of 0.1 µg/mL. After washing, the immunoreactive bands were visualized using ECL Prime Western Blotting Detection Reagent (GE Healthcare, Tokyo, Japan), and the band intensities were measured with a ChemiDoc™ MP Imaging System (Bio-Rad Laboratories, Hercules, CA, USA).

### 2.4. Quantitative Real-Time PCR (qPCR) Analysis

Total RNA was extracted from 20 mg rat soleus samples stored with RNAlater (Thermo Fisher Scientific) at −20 °C using a TissueLyser and RNeasy Mini Kit with DNase (Qiagen, Valencia, CA, USA). qPCR was performed using an ABI PRISM 7500 Fast Real-Time PCR system (Thermo Fisher Scientific K.K., Uppsala, Sweden) with SYBR Premix Ex Taq (TaKaRa Bio, Shiga, Japan) following reverse transcription of the RNA into cDNA using a PrimeScript™ RT Reagent Kit (TaKaRa Bio). The primer sets used in this study are shown in Appendix A. The expression levels of the target mRNAs were normalized to those of GAPDH mRNA.

### 2.5. Histological Analysis

After scarification, the right gastrocnemius of rats was fixed in 4% paraformaldehyde (Wako, Tokyo, Japan) at 4 °C for 2 days and then placed in 70% ethanol for storage. The thickest part of the fixed tissues was cut out, embedded in paraffin block, and sliced to 3 µm using a microtome. The sliced tissues were adhered to silane-coated slide glass (Muto Pure Chemicals Co., Ltd., Tokyo, Japan), which was used for immunohistochemistry (IHC) against SERCA2 ATPase. The deparaffinizing tissues with xylene and ethanol were incubated in Histo VT One (Nacalai tesque, Inc., Kyoto, Japan), heated antigen retrieval solution for 30 min at 90 °C, and reacted with the first antibody against SERCA2 ATPase (ab2861, Abcam, Cambridge, UK) overnight at 4 °C after blocking with 5% normal goat serum (Nichirei bioscience, Inc., Tokyo, Japan) for 1 h at room temperature (RT). After washing with PBS, the slides were reacted with Histofine Simple Stain Rat MAX-PO (M) (Nacalai tesque, Inc.) for 30 min at RT. After washing with Tris-buffered saline (TBS) buffer, ImmPACT™ DAB (Vector Laboratories, Inc., Burlingame, CA, USA) and hematoxylin were used for staining. The 450 fibers in a medial head of gastrocnemius were randomly counted and SERCA2 ATPase positively stained fibers were viewed using a stereoscopic microscope BX53 and Olympus cellSens Dimension software (Olympus, Tokyo, Japan). This counting was carried out under the blind. The images are provided in Appendix A.

### 2.6. Grip Test

The fitness performance of the treated mice was determined at 2 and 4 weeks by testing whole-body grip strength using 4 limbs with a grip strength machine (MELQUEST Co., Ltd., Toyama, Japan), as described previously with slight modifications [19]. The grip tests were repeated five times, and all replicates were conducted within 5 min. The median was used as a representative value.

### 2.7. Statistical Analysis

The data are presented as means and standard errors (SEs). One-way ANOVA was used for parametric analyses, and followed by a Student’s *t*-test or Tukey–Kramer test. The Mann–Whitney U test and Steel–Dwass test were used as the non-parametric analysis methods.

### 2.8. Data Availability

The datasets generated and analyzed during the current study are available from the corresponding author, K.T., upon reasonable request.

## 3. Results

To verify the beneficial effect of B-3 administration on muscle tissue and muscle function, we used two different animals. Rats were used to examine the muscle mass and some activated signaling pathways (Experiment 1). On the other hand, mice were used for functional evaluation of B-3HK on the muscle strength (Experiment 2).


**Experiment 1**


### 3.1. The Effects of B-3 Administration on Body and Tissue Weight in Rats

We first evaluated changes in muscle mass in rats after B-3 administration according to the schedules described (Figure 1). A significant increase in the weight of the soleus (a slow oxidative (SO) fiber-dominant muscle) per total body weight was observed in the B-3HK group compared with the control group. In addition, the weights of the plantaris and gastrocnemius (fast glycolytic (FG) fiber-dominant muscles) per total body weight tended to be higher in the B-3HK group than in the control group (Table 1), but without statistical significance. Although the masses of these muscles were also higher in the B-3L group and the leucine (a muscle hypertrophy promoting nutrients) group than in the control group, the differences were not significant. A significant difference in food intake was observed at week 1 between the control and B-3HK groups, but there were no significant differences afterwards. There were no significant differences in body weight at each time point, or in liver weights at week 4, between each group (Table 1).

### 3.2. B-3HK Promoted Phosphorylation of Akt in the Rat Soleus

To clarify the mechanisms by which B-3HK increased muscle mass, we investigated the activation of Akt, one of the key regulators of protein synthesis in skeletal muscle [20]. The western blot results indicate that the levels of activated Akt in the soleus were significantly increased by B-3HK administration (Figure 2). It is well known that activated Akt in skeletal muscle induces the mTOR signaling pathway, which is a critical pathway regulating protein synthesis, leading to increased muscle mass. We observed that B-3HK tended to induce phosphorylation of mTOR and its downstream molecule p70S6K in soleus, similarly to leucine (Figure 3), indicating that the possible involvement of the Akt-mTOR-p70S6K signaling pathway in increased muscle mass by B-3HK administration.

### 3.3. The Effects of B-3 on the AMPK-PGC-1α-Mitochondrial Biogenesis Pathway in the Rat Soleus

We then examined the effects of B-3 administration on muscle metabolism, which is related to energy productivity and consumption [21]. B-3HK significantly promoted the activation of AMPK (Figure 4A,B), which regulates energy balance at both the cellular and physiological levels as a master regulator of metabolism [22,23]. No change was observed in the group treated with leucine, a well-known muscle hypertrophy-promoting nutrient. Moreover, B-3HK also significantly increased the mRNA expression of peroxisome proliferator-activated receptor gamma coactivator (PGC)-1α, a master regulator of mitochondrial biogenesis [24], and cytochrome *c* oxidase (CCO), which has been reported to control enzymes related to the mitochondrial oxidative phosphorylation system (OXPHOS) at the mitochondrial membrane [25,26], in the soleus (Figure 4C,D). Although the expression of these genes was also higher in the B-3L group than in the control group, the differences were not significant.

### 3.4. B-3HK Promoted the Distribution of Oxidative Fibers in the Gastrocnemius in Rats

A high expression of PGC-1α is well known to be involved in shifting muscle fiber type distribution toward oxidative fibers [27]. To examine the enhancing effect of B-3HK on the oxidative fiber composition in the gastrocnemius, an FG fiber-dominant muscle adjoining soleus was evaluated by counting the number of sarco-/endoplasmic reticulum Ca^2+^ (SERCA)2 ATPase-positive fibers (Figure 5A). The administration of B-3HK significantly increased the relative abundance of type I muscle fibers (the most oxidative fibers) compared with saline administration (Figure 5B). B-3HK had a greater effect than B-3L in promoting the distribution of oxidative fibers. This finding indicates that B-3HK induced PGC-1α expression in the gastrocnemius as well as in the soleus.


**Experiment 2**


### 3.5. B-3HK-Enhanced Fitness Performance in Mice

Since increasing muscle mass was expected to enhance muscle strength, we used a murine model to evaluate the effects of B-3HK on muscle function. We observed a significantly higher level of grip strength in the B-3HK group compared to the control group at both two and four weeks (Figure 6), in addition to significantly increased muscle mass in the mice’s solei (Appendix A).

## 4. Discussion

Skeletal muscle is the largest organ in the body; in most mammals, it makes up ~45%–55% of the body mass and is a major determinant of the basal metabolic rate [28,29,30]. Our previous clinical results in mild obesity subjects indicated that B-3 administration both decreases body fat percentage and increases muscle mass [2]. Thus, we hypothesized that B-3 exerts beneficial effects on muscle mass and function related to basal muscle metabolism that might contribute to its anti-obesity effects.

Indeed, in the present study, we found that daily administration of B-3 to rats for a prolonged period tended to increase muscle mass and activate AMPK, a well-known master metabolic regulator, in the soleus [22,23]. Furthermore, the gene expression of PGC-1α and CCO was higher in the B-3 administration group than in the control group. The increase in PGC-1α and CCO gene expression indicates a possibility of the enhancement of basal metabolism through mitochondrial biogenesis and OXPHOS [24,25,26]. Thus, these results suggest that B-3 induces mitochondrial biogenesis in skeletal muscle through the AMPK-PGC-1α signaling pathway, which might be a mechanism by which B-3 exerts its anti-obesity effects.

Intriguingly, B-3HK had a greater effect than B-3L in eliciting some muscle improvements. For example, B-3HK activated Akt-mTOR-protein synthesis signaling, and consequently increased muscle mass. Through its beneficial effects, B-3HK was observed for the possible enhancement of grip strength in a murine model. In addition, B-3HK was suggested to enhance mitochondrial biogenesis (e.g., increase CCO gene expression) through AMPK-PGC-1α signaling pathways, probably leading to increased mitochondrial energy productivity, and induced an oxidative fiber type composition in the gastrocnemius.

In general, AMPK is well known to inhibit protein synthesis through the suppression of the mTOR, a critical regulator of anabolic pathways [22,23], which indicates that the co-activation of both anabolic and catabolic pathways does not occur in the skeletal muscle at the same time. However, we observed the co-activation of Akt and AMPK in response to B-3HK by western blotting analysis. These results suggest that several independent pathways from B-3HK might activate both Akt and AMPK. Since the bacterial cells of B-3HK might contain various components, such as lipoteichoic acid, peptide glycans and nucleotides, each component might separately activate the Akt and/or AMPK signaling pathway with various strengths in the skeletal muscle. As a result, the mTOR-protein synthesis pathway might be positively influenced in response to B-3HK for muscle hypertrophy, in spite of AMPK activation. To unravel the reasons for this, further studies are required for the identification of bioactive factors.

Similarly to our observation, it has been reported that *Lactobacillus plantarum* TWK10 increases muscle mass and promotes type I fibers (slow muscle) in murine gastrocnemius muscle [31]. Moreover, TWK10 has shown potential as an ergogenic aid to improve aerobic endurance performance via physiological adaptation effects in amateur runners and healthy humans [32,33].

Furthermore, the way in which heat-killing treatment enhances the effects of B-3 remains to be elucidated. *Plovier* et al. reported that heat-killed *Akkermansia muciniphila* exerted stronger anti-obesity effects than live *A. muciniphila* in high-fat diet-fed mice [34,35]. In addition, another study reported that although the administration of heat-killed *A. muciniphila* improved several metabolic parameters, such as insulin sensitivity and plasma total cholesterol, in obese insulin-resistant volunteers, it did not change the gut microbiome structure [36]. Therefore, it has been suggested that the mechanism by which heat-killed *A. muciniphila* exerts its effects involves the modulation of host immune responses and the intestinal barrier function by an outer membrane protein (Amuc_1100 protein) through activation of TLR2 and TLR4 [34,35]. It is also well known that *Bifidobacterium* has cell components that can regulate TLR signaling pathways [37,38]. Thus, we speculate that certain cell component(s) of B-3HK might exhibit enhanced ligand binding with targeted receptors to activate signaling pathways in intestinal cells (e.g., intestinal epithelial cells or intestinal immune cells), and then induce the production of some bioactive factors. Notably, the administration of serum from B-3HK-treated rats activated AMPK in a differentiated L6 rat myoblast cell line (Appendix A), indicating that bioactive factors such as cytokines and hormones could have been present in the serum. In addition, some reports suggested that live and heat-killed probiotics strains had different immune responses, such as cytokine secretion profiles [39,40], which may be one of the possible mechanisms by which heat-killed B-3 is more effective on muscle functions. To clarify the differences in activity between B-3L and B-3HK, further studies are needed to elucidate the specific B-3HK-affected signaling pathway and to identify the bioactive factors in B-3HK-treated rat serum.

Recent studies have suggested that *Bifidobacterium* is recognized as a key taxon for physical frailty and sarcopenia in elderly individuals [8]; however, it remains unclear whether the decreases in bifidobacterial abundance that occur with age influence muscle health [7,41]. The present study explored the positive relationships between *Bifidobacterium* and muscle health. Although further studies using ageing models [42] are required, B-3HK was expected to be useful for improving muscle atrophy, such as that occurring in sarcopenia according to the increase in muscle mass by B-3HK.

There are some limitations in this study. Firstly, we used leucine as a positive control for activation of mTOR signaling pathway. Leucine tended to promote phosphorylation mTOR and p70S6K (Figure 3b,c), but the effect appeared to be weaker than expected. Moreover, owing to the limits of administrated volume and the solubility of leucine in saline, leucine was administrated at 0.131 g/kg/day, which was lower compared to previous study (>0.675 g/kg/day) by Fumiaki et al. [43]. The low dosage of leucine used in the present study might have influenced the degree of mTOR phosphorylation induction. Secondly, the mechanisms of the effects of B-3, and the reason why heat-killed B-3 was more effective, were not clarified. Moreover, since the results were observed in animal models, translation of the effects to humans remains unknown. These issues need to be addressed in future studies.

## 5. Conclusions

Daily administration of B-3, a probiotic strain for anti-obesity, promoted mitochondrial biogenesis through the AMPK-PGC-1α signaling pathway in skeletal muscle, and the distribution of oxidative muscle fibers through increasing in PGC-1α gene expression; and induced muscle hypertrophy through the Akt signaling pathway in rats. Remarkably, it was observed that heat-killing treatment enhanced the activities of B-3. In addition, B-3HK also showed potential in increasing grip strength in a mouse model. These findings may contribute to the understanding of the mechanism of the anti-obesity effects of B-3 and suggest its potential benefits for improving physical fitness and ameliorating physical fatigue and muscle atrophy; in particular, after heat-killing treatment.

## Figures and Tables

**Figure 1 nutrients-12-00219-f001:**
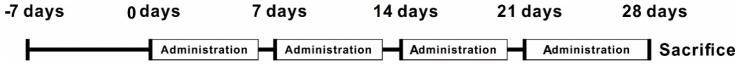
Protocol for the In Vivo test. Fifty male rats were acclimatized to the environmental conditions for 7 d after receipt and were then divided into four groups (*n* = 12 or 13). Each treatment (e.g., B-3L or B-3HK) was orally administered for 4 weeks. Body weight and food intake were monitored weekly. After administration for 4 weeks, the tissues were collected in order to evaluate the effects on muscle.

**Figure 2 nutrients-12-00219-f002:**
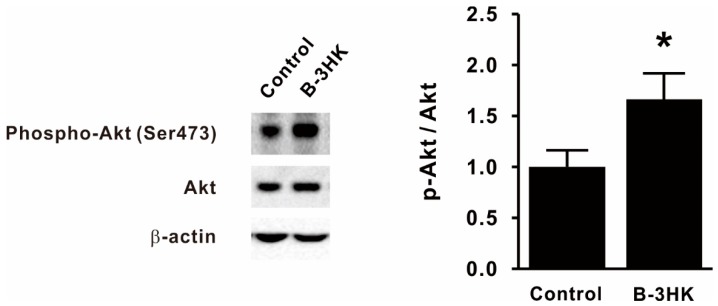
The effects of B-3 on Akt activation in the soleus. Representative bands and the expression ratio of pAkt to Akt in the soleus are shown (*n* = 9, 10). Difference was analyzed for statistical significance using Mann–Whitney U test (* *p* < 0.05). The data are expressed as the means and SEs.

**Figure 3 nutrients-12-00219-f003:**
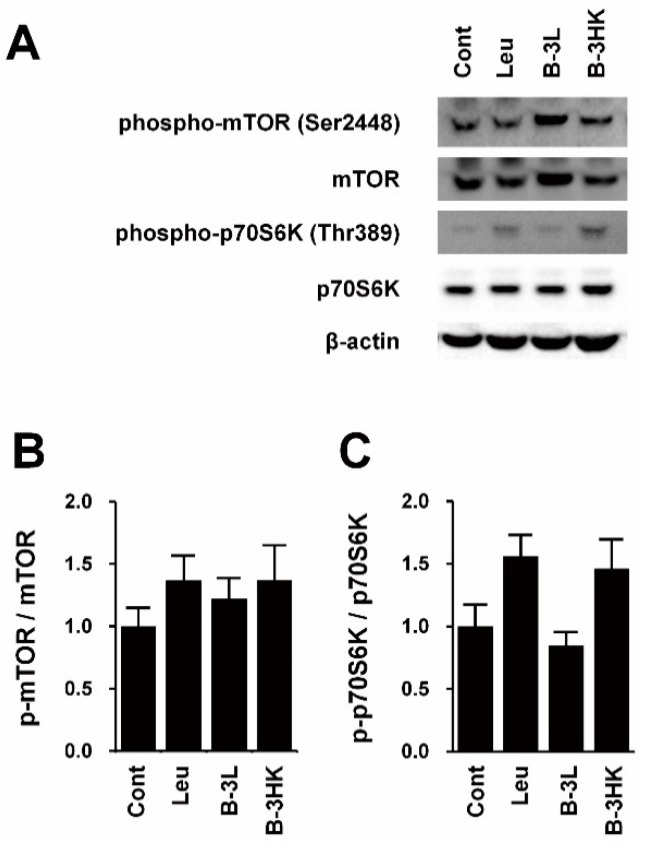
The effects of B-3 on mTOR signaling in soleus. Western blotting was used for the signaling pathways related protein synthesis in rat soleus muscle. (**A**) Representative bands and the expression ratio of (**B**) p-mTOR to mTOR and (**C**) p-p70S6K to p70S6K are shown (*n* = 6). Leucine was used as a positive control for induction of mTOR. The data are expressed as the means and SEs. Abbreviations: Cont, control; Leu, leucine; B-3L, live B-3; B-3HK, heat-killed B-3.

**Figure 4 nutrients-12-00219-f004:**
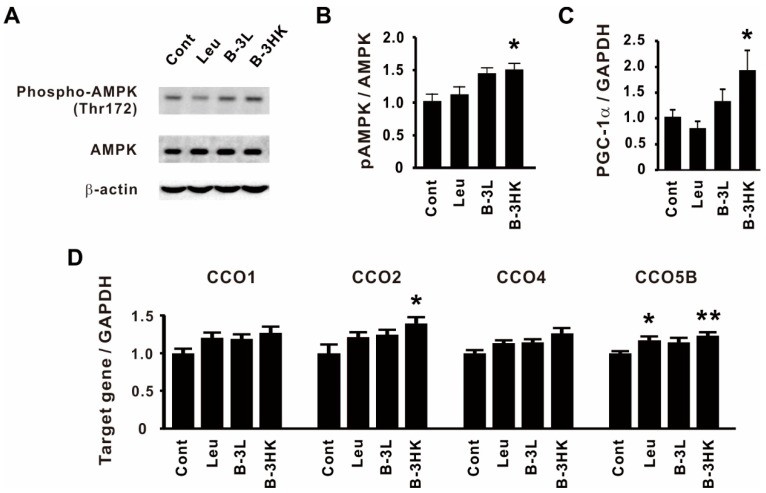
B-3 enhanced the AMPK-PGC-1α-mitochondrial biogenesis signaling pathway in the soleus. (**A**) Representative bands and (**B**) the expression ratio of pAMPK to AMPK in the soleus are shown (*n* = 6). (**C**,**D**) qPCR was used to evaluate the relative mRNA expressions of PGC-1α, CCO1, CCO2, CCO4 and CCO5B in the soleus (*n* = 12–13). Statistical differences were analyzed for significance using the Steel–Dwass test (* *p* < 0.05, ** *p* < 0.01, versus control). The data are expressed as means and SEs. Abbreviations: Cont, control; Leu, leucine; B-3L, live B-3; B-3HK, heat-killed B-3.

**Figure 5 nutrients-12-00219-f005:**
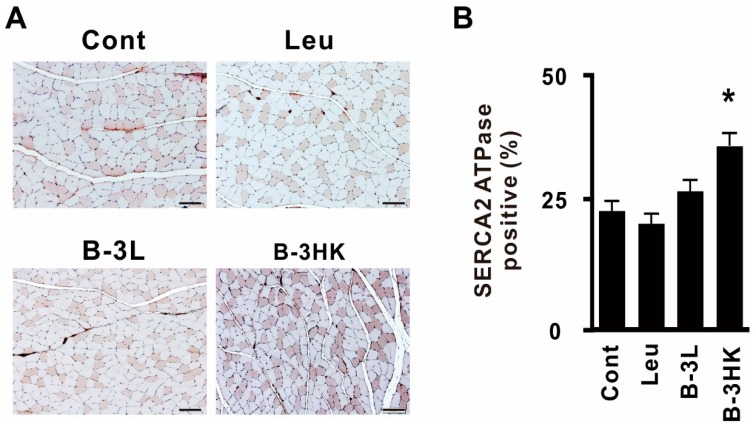
B-3 promoted the switching of type 1 fibers through the AMPK signaling pathway in rats. (**A**) Representative results of immunohistochemistry against SERCA2 ATPase in the rat medial gastrocnemius. The scale bar represents 100 µm. (**B**) The mean percentage of SERCA2 ATPase-positive fibers was determined among 450 fibers in three sections per rat (*n* = 7). Statistical differences were analyzed for significance using the Steel–Dwass test (* *p* < 0.05, versus control). The data are expressed as the means and SEs.

**Figure 6 nutrients-12-00219-f006:**
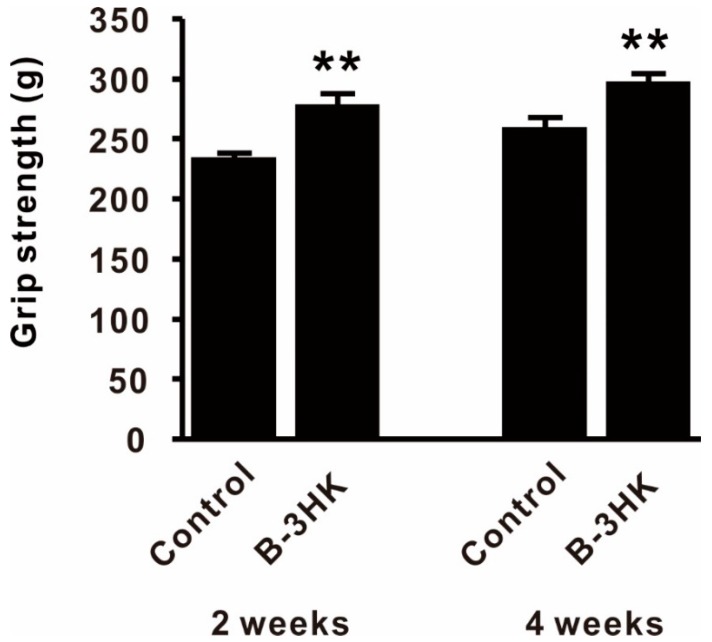
The effect of B-3 on whole-body grip strength in mice. Physical performance was analyzed by the grip strength test at 2 and 4 weeks (*n* = 18). Differences with the control group were analyzed for significance using Student’s *t*-test (** *p* < 0.01). The data are expressed as the means and SEs.

**Table 1 nutrients-12-00219-t001:** The effects of B-3 on body and tissue weights in rats.

		Control	Leucine	B-3L	B-3HK
Body weight (g)	initial	311.2 ± 2.7	307.3 ± 3.5	302.1 ± 2.9	303.8 ± 3.6
	1 week	361.1 ± 4.6	356.9 ± 5.5	358.4 ± 3.3	354.1 ± 5.1
	2 weeks	408.0 ± 6.3	397.3 ± 9.5	400.5 ± 4.8	393.2 ± 7.5
	3 weeks	430.7 ± 9.6	418.1 ± 9.7	425.9 ± 6.5	417.9 ± 9.0
	4 weeks	433.2 ± 9.4	430.6 ± 10.7	430.1 ± 6.2	421.3 ± 9.4
Food intake (g)	initial	31.8 ± 0.6	30.7 ± 0.8	30.5 ± 0.5	31.0 ± 0.9
	1 week	33.7 ± 1.1	31.6 ± 1.3	31.4 ± 0.7	29.8 ± 1.1 *
	2 weeks	31.2 ± 0.9	30.3 ± 1.0	30.7 ± 1.5	29.5 ± 1.1
	3 weeks	30.8 ± 1.1	28.5 ± 1.2	30.7 ± 1.5	30.0 ± 1.3
	4 weeks	33.5 ± 1.2	32.1 ± 1.4	31.8 ± 0.7	32.2 ± 1.1
Liver weight (g)		12.0 ± 0.4	11.5 ± 0.4	11.7 ± 0.3	11.3 ± 0.4
Liver weight/body weight (mg/g)		27.6 ± 0.8	26.5 ± 0.5	27.2 ± 0.5	26.8 ± 0.4
Soleus weight (mg)		193.3 ± 5.8	203.5 ± 5.7	205.4 ± 5.7	209.7 ± 5.4
Soleus weight/body weight (mg/g)		0.45 ± 0.01	0.47 ± 0.01	0.48 ± 0.01	0.50 ± 0.01 *
Plantaris weight (mg)		415.0 ± 21.6	431.4 ± 12.4	440.3 ± 12.4	456.4 ± 24.4
Plantaris weight/body weight (mg/g)		0.96 ± 0.05	1.00 ± 0.02	1.03 ± 0.03	1.08 ± 0.05
Gastrocnemius weight (mg)		2242.9 ± 50.3	2253.3 ± 48.1	2303.5 ± 44.8	2274.2 ± 53.4
Gastrocnemius weight/body weight (mg/g)		5.19 ± 0.09	5.26 ± 0.13	5.37 ± 0.12	5.40 ± 0.06

The data are presented as the means and SEs. Asterisks (*) denote significant differences from the control group at * *p* < 0.05 (Tukey–Kramer test).

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
