# Peer review of "Heat-Killed Bifidobacterium breve B-3 Enhances Muscle Functions: Possible Involvement of Increases in Muscle Mass and Mitochondrial Biogenesis"

_nutrients, 2020, doi:10.3390/nu12010219_

Round 1

Reviewer 1 Report

General:

The manuscript “Heat-killed Bifidobacterium breve B-3 enhances muscle functions: Possible involvement of increases in muscle mass and mitochondrial biogenesis” by Toda et al. focus on potential role of B-3 as a muscle hypertrophy-inducing agent. It’s an interesting study, but the manuscript also needs further work improving the writing. Since there are 2 experimental designs in this study, experiment 1 uses rats and experiment 2 uses mice. The authors need to explain the purpose of using 2 different animals in different experiments. In addition, the part of the result should also be divided into two parts, and the results of Experiment 1 and Experiment 2 are presented in sequence. This will make the readers easy to read.

Specific:

Material and methods

Line 65: The preparation of heat-killed B-3 is not clear. And the citation Ref.#21 did not show the heat-killed B-3. Authors need to explain this part. Line 80: Please describe what kind of treatment on day 6th, 13th and 20th.

Results:

Table 1: There is a significant difference of food intake at week 1st, between control and B-3HK groups. Is there any explanation? Table 1: In addition to relative weights of different tissues, I suggest authors to provide the actual weights of liver, soleus, plantaris and gastrocnemius in the table 1.

Discussion:

I suggest authors could discuss their findings more extensive and include two important articles into this manuscript. Chen, Y.M.; Wei, L.; Chiu, Y.S.; Hsu, Y.J.; Tsai, T.Y.; Wang, M.F.; Huang, C.C. Lactobacillus plantarum TWK10 Supplementation Improves Exercise Performance and Increases Muscle Mass in Mice. Nutrients 2016, 8, 205. Huang, W.C.; Lee, M.C.; Lee, C.C.; Ng, K.S.; Hsu, Y.J.; Tsai, T.Y.; Young, S.L.; Lin, J.S.; Huang, C.C. Effect of Lactobacillus plantarum TWK10 on Exercise Physiological Adaptation, Performance, and Body Composition in Healthy Humans. Nutrients 2019, 11, 2836. The dosage is always an important issue to convert the treatment of rodents to human. Authors need to describe how much dosage of B-3HK is suggested for healthy adults.

Reviewer 2 Report

The subject of the article is interesting, the review, material and methods as well as the results presented in an accessible way.
A large number of results apresented as a supplement, but I would like to see histological pictures from muscle staining. Those attached to the article cannot be enlarged. Were the fiber diameters measured and counted?

Reviewer 3 Report

The authors set out to determine the effects of 4-weeks of either live or heat-killed B-3 on muscle function (i.e. muscle strength and metabolisms) and signaling pathways controlling muscle hypertrophy in rodent skeletal muscle. Increasing effects on muscle function (grip strength), muscle mass, and muscle metabolism were observed with B-3 supplementation, and even stronger effects were seen in the B-3HK administrated animals. In addition, the activation of both Akt and AMPK-PGC1α-mitochondrial biogenesis pathway were observed when the rodents were provided with B-3HK. The study protocol and methodology are generally well conceived. However, there were lacking of thoroughly discussion about that the signaling pathways governing both catabolism (AMPK) and anabolism (Akt) were dually activated in response to B-3HK administration. Moreover, there are some more specific concerns listed below.  

Specific concerns:

Lack of familiarity with English sentence structure and usage leads to several cases of contradicting yourself in the same paragraph as well as questions as to how accurately some previous work was described. The paper needs to be gone over by someone familiar with the area of work who is highly fluent in English. The signaling pathways selected to detected should be justified in the introduction. Please address. AMPK plays as the upstream inhibitory signal of mTOR, which is the key signal controlling muscle growth, whereas this key signal molecule was not detected and discussed. Moreover, for muscle hypertrophy, mTOR is a more predominate signals governing skeletal muscle protein accretion and hypertrophy. Therefore, if possible it would be more appropriate to add experiment to identify the role of mTOR in the benefits of B-3HK administration. Please address. The authors used the upper limb grip strength as the muscle performance and selected to analyze the soleus/gastrocnemius for determining muscle metabolic related signaling. However, there were lack of discussion and cited references to provide the possible linkage for assessing the muscle function and related signaling using muscle from different limbs. Please address. Further discussion is warranted as to why and the underlying mechanism for which the heat-killed B-3 would generate greater muscle function/metabolic benefits, which would be important to provided possible explanation for the different responses between live and heat-killed B-3 in this study. A paragraph of practical application and limitation must be added in the last part of the discussion to provide translational/application information for readers.

Round 2

Reviewer 1 Report

Authors have been responded to every comment positively, but I still have additional suggestions as following:

1. The data in Fig. 2. Fig. 3, Fig. 4, and Supplementary Figs. S2 and S3 are non-parametric data, however, they used t-test and Tukey-Kramer test (for parametric data).

2. Authors should have used the non-parametric analysis methods (such as Mann-Whitne U-test for 2 groups comparison, and Kruskal-Wallis test post hoc Dunn test for multiple comparison (or Steel Dwass test).

3. There is no group Leucine in the supplementary figure 1 (a)-(c), and the quantity of beta-actin (internal control) is not equally in each sample. I suggest authors delete this data at the published stage.

Author Response

Thank you for your advice. The statistical analysis in the applicable figures was changed into the non-parametric analysis methods (Mann-Whitne U-test and Steel Dwass test). Furthermore, following your 3rd advice, we removed the supplemental figure 1.

Reviewer 3 Report

Thank you for all the efforts the authors made to respond my comments, and the questions or comments provided have already been addressed and revised in the resubmitted version.

However, I am still considering that the mTOR-related signaling results should be appeared in the results section instead of presenting in the supplementary data (Supplemental Figure 3), because these signal proteins are more critical and direct molecules controlling muscle protein synthesis. Moreover, it has to be noted that leucine has been demonstrated as a strong stimulant for mTOR phosphorylation, and this should be the reason why author used this treatment as the positive control. But, only very little activation of this protein-synthetic controlling protein was detected in this study (Supplemental Figure 3; it was even weaker than B-3L treatment), although its downstream molecule, p70S6K, exhibited clear activation. I will suggest the author to provide a reasonable explanation about this distinct finding in the discussion section.

For this change, all the corresponding content and descriptions in the maintext should be modified and revised after re-ordering the results presentation. If all the changes and related explanation in the results/discussion have been made, do a excellent work.

Author Response

Thank you very much for kindly reviewing again. As you mentioned, we used leucine as a positive control for activation of mTOR signaling pathway. Leucine tended to promote phosphorylation not only of p70S6K but also of mTOR (Figure 3b and 3c). However, as you pointed out, the effect of leucine on activating mTOR in this study was weaker than expected.

We matched the dosage volume of leucine dissolved in saline with the B-3 administration groups. We recalculated leucine intake and found that leucine was administrated at 0.131g/kg/day, which was lower compared to previous study (>0.675g/kg/day, Fumiaki et al (Biosci Biotechnol Biochem, 77 (4), 839-842, 2013)).

The low dosage of leucine used in the present study might have influenced the degree of mTOR phosphorylation induction. Thus, we thought leucine was inadequate as positive control in this study. If you give us more time, we will retest and change the figures and text to remove the data with leucine group.

This manuscript is a resubmission of an earlier submission. The following is a list of the peer review reports and author responses from that submission.

Round 1

Reviewer 1 Report

The reviewed manuscript by Toda and colleagues seeks to advance understanding of the gut-muscle axis by evaluating the efficacy of a 4-wk Bifidobacterium breve B-3 (B-3) supplement on skeletal muscle health in male rodents. Using two independent experimental models, the authors utilize a diverse array of experiments in an attempt to examine the influence of the supplement from the whole muscle (size and function) to the gene level. While the presented results portend to the robust myogenic potential of the supplement, unfortunately, I have serious concerns with the employed statistical approach and data interpretation. These issues are outlines below.

Major Comments:

Statistical Analysis – by far my largest concern has to do with the employed statistical techniques. Were the data checked for normality? If not, Student’s t-tests and Dunnett’s assessment of multiple comparisons cannot be utilized. In figure 2B the authors use a Spearman correlation, implying were not normally distributed. Moreover, the use of Dunnett’s multiple comparison tests (i.e., many-to-one comparison) is inappropriate for the present analysis, leading to misinterpretation of findings. This statistical technique compares all of the experimental groups as a whole to the control group, and thus is particularly prone to Type I error. Moreover, Dunnett’s test does not facilitate discrimination among the experimental groups. Thus, using this test it cannot be discerned how the authors were able to make claims such as those in Results lines 136-138, stating that soleus weight was greater in the B-3HK than the control group. Moreover, the authors follow this statement by claiming relative gastroc/plantaris weights were also greater in the B-3HK group yet statistical significance was not achieved.

Introduction – please elaborate as to why heat-killed B-3 were administered. It would seem the majority of the beneficial effects of bacterial administration would occur from colonization/transformation of gut microbiota composition.

Methods – was the heat-killing protocol a fabrication of the authors or has this been used before? Given the novelty of the technique a standardized heat-killing protocol is desirable.

Methods/Results – the research teams objective to uncover the molecular determinants of possible skeletal muscle changes is appreciated, but it is somewhat unclear as to why the majority of the chosen gene targets (AMPK, PGC1a, etc.) are related to mitochondrial biogenesis and oxidative metabolism rather than muscle growth and gains in strength, which were the primary focus of the current analyses.

Methods/Results – why was fiber type assessed via SERCA2 quantification rather than more conventional mATPase histochemistry or immunohistochemistry?

Results – A major claim of the paper is that B-3HK administration “increased” muscle strength. Yet, the experimental technique does not support this assertion. Was muscle strength evaluated prior to B-3HK administration (i.e., baseline)? Baseline data must be presented to delineate that muscle strength was actually “improved” and not simply that it was always greater in the treatment group.

Author Response

Response to Reviewer 1 Comments

Comments and Suggestions for Authors

The reviewed manuscript by Toda and colleagues seeks to advance understanding of the gut-muscle axis by evaluating the efficacy of a 4-wk Bifidobacterium breve B-3 (B-3) supplement on skeletal muscle health in male rodents. Using two independent experimental models, the authors utilize a diverse array of experiments in an attempt to examine the influence of the supplement from the whole muscle (size and function) to the gene level. While the presented results portend to the robust myogenic potential of the supplement, unfortunately, I have serious concerns with the employed statistical approach and data interpretation. These issues are outlines below.

We appreciate the time and effort you and each of the reviewers have dedicated to providing insightful feedback on ways to strengthen our paper. Thus, it is with great pleasure that we resubmit our article for further consideration. We have incorporated changes that reflect the detailed suggestions you have graciously provided. We also hope that our edits and the responses we provide below satisfactorily address all the issues and concerns you and the reviewers have noted.

To facilitate your review of our revisions, the following is a point-by-point response to the questions and comments delivered in your letter.

Major Comments:

Statistical Analysis – by far my largest concern has to do with the employed statistical techniques. Were the data checked for normality? If not, Student’s t-tests and Dunnett’s assessment of multiple comparisons cannot be utilized. In figure 2B the authors use a Spearman correlation, implying were not normally distributed. Moreover, the use of Dunnett’s multiple comparison tests (i.e., many-to-one comparison) is inappropriate for the present analysis, leading to misinterpretation of findings. This statistical technique compares all of the experimental groups as a whole to the control group, and thus is particularly prone to Type I error. Moreover, Dunnett’s test does not facilitate discrimination among the experimental groups. Thus, using this test it cannot be discerned how the authors were able to make claims such as those in Results lines 136-138, stating that soleus weight was greater in the B-3HK than the control group. Moreover, the authors follow this statement by claiming relative gastroc/plantaris weights were also greater in the B-3HK group yet statistical significance was not achieved.

Thank you very much for detail checks. All data were rechecked for normality, and were confirmed for no problems in according to histogram. The statistical analysis in the figure 2B was changed into the Pearson's correlation coefficient as parametric test (p.6, lines 164). Moreover, this study aimed to evaluate the beneficial effects of each sample rather than to compare the efficacy of each sample, for which we think a many-to-one comparison test could be more suitable for the purpose. Similar to our study, Dunnett’s multiple comparison tests were also used in similar researches in the evaluation of beneficial effects probiotics [Nat Med. 2017, 23(1), 107-113., PLoS One. 2014, 9(3), e91857., Nutrients. 2018, 10(5), pii: E643]. In addition, since similar results were observed in an independent reproducibility test (Supplementary data in Table 2 and Figure 2), we thought these effects were not accidentally observed. About the effects on gastroc/plantaris weights, we have rephrased (p.6 , lines 147-149) to avoid the over-speculation.

Introduction – please elaborate as to why heat-killed B-3 were administered. It would seem the majority of the beneficial effects of bacterial administration would occur from colonization/transformation of gut microbiota composition.

Thank you for your suggestions. We have added and rephrased the sentences (p.1-2, lines 39-50).

Methods – was the heat-killing protocol a fabrication of the authors or has this been used before? Given the novelty of the technique a standardized heat-killing protocol is desirable.

Description of heat-killing method was added (p.2, lines 54-55).

Methods/Results – the research teams objective to uncover the molecular determinants of possible skeletal muscle changes is appreciated, but it is somewhat unclear as to why the majority of the chosen gene targets (AMPK, PGC1a, etc.) are related to mitochondrial biogenesis and oxidative metabolism rather than muscle growth and gains in strength, which were the primary focus of the current analyses.

Thank you for your suggestion. Since some sections were likely to cause misunderstanding, the title and text were rephrased (p.9, lines 2329-234) to better clarify our meaning. We also think that increases in oxidative capacity might not directly contribute with enhancement of grip strength.

Methods/Results – why was fiber type assessed via SERCA2 quantification rather than more conventional mATPase histochemistry or immunohistochemistry?

As you may know, SERCA2 is referred to ATPase sarcoplasmic/endoplasmic reticulum Ca2+ transporting 2 (ATP2A2) or calcium-transporting ATPase sarcoplasmic reticulum type / slow twitch skeletal muscle isoform according to NIH database; https://ghr.nlm.nih.gov/gene/ATP2A2#sourcesforpage. Moreover, SERCA2a is located primarily in heart and slow-twitch skeletal muscle, whereas SERCA2b is present in smooth muscle and nonmuscle tissues [Pharmacol Ther. 1991, 50(2), 191]. Therefore, we thought SERCA2 was able to use for an indicator of slow-twitch muscle fibers in skeletal muscle.

Results – A major claim of the paper is that B-3HK administration “increased” muscle strength. Yet, the experimental technique does not support this assertion. Was muscle strength evaluated prior to B-3HK administration (i.e., baseline)? Baseline data must be presented to delineate that muscle strength was actually “improved” and not simply that it was always greater in the treatment group. 

As you mentioned, the exactly evaluation of grip strength might need baseline. In this study, the mice were randomly grouped based on body weights, and body weight is generally understood to correlate muscle mass in healthy young mice. In fact, the evaluation of muscle weights was usually normalized with body weight and some studies also evaluated murine grip strength in only endpoint without initial, as well as this study [e.g., Nutrients 2016, 8(10), 648., Nutrients 2016, 8(4), 205.]. Therefore, we think that the data for grip strength obtained in this study is reasonable. In addition, similar results were observed in an independent reproducibility test, which suggested that B-3HK enhanced the murine grip strength.

Again, thank you for giving us the opportunity to strengthen our manuscript with your valuable comments and queries. We have worked hard to incorporate your feedback and hope that these revisions persuade you to accept our submission.

Sincerely,

Kazuya Toda.

Reviewer 2 Report

The current paper has investigated the effects of a probiotic strain on muscle mass, function and gene expression in rodents. There appears to be a positive effect of heat killed strains, although the mechanism by which this appears to be more beneficial than live bacteria is not elucidated.

Introduction

Very clear and succinct introduction. Would you be able to elaborate slightly though on the pathway of mechanism by which probiotics affecting the GI tract would then impact skeletal muscle physiology and functionality. You could also elaborate on the reasoning for adding a heat treated arm of the study.

Methods

Line 61-62 - how was food intake monitored?

Results

Table 1 - it is not clear what the P value represents? If all three conditions were compared to Control, for which does the P value represent? If any condition is significantly different to control, could you not indicate that in the relevant column?

Line 135 - 143 - could you provide individual p values where you describe differences to control group. You have described that plantaris and gastrocnemius relative weights increased, yet the table shows p > 0.05, suggesting no significant difference.

Discussion

Line 221 - I think it would be more accurate to describe that Akt signalling was associated with increases in soleus mass.

Line 222 - You did not assess mitochondrial biogenesis, so can only speculate that it would have been enhanced, along with mitochondrial energy production.

224 - Again, while you can speculate that these adaptations are the cause of the increase in grip strength, you can only speculate on this association. However, I believe you need to make it clearer how increases in oxidative capacity would benefit muscular strength? 

Author Response

Response to Reviewer 2 Comments

Comments and Suggestions for Authors

The current paper has investigated the effects of a probiotic strain on muscle mass, function and gene expression in rodents. There appears to be a positive effect of heat killed strains, although the mechanism by which this appears to be more beneficial than live bacteria is not elucidated.

We appreciate the time and effort you and each of the reviewers have dedicated to providing insightful feedback on ways to strengthen our paper. Thus, it is with great pleasure that we resubmit our article for further consideration. We have incorporated changes that reflect the detailed suggestions you have graciously provided. We also hope that our edits and the responses we provide below satisfactorily address all the issues and concerns you and the reviewers have noted.

To facilitate your review of our revisions, the following is a point-by-point response to the questions and comments delivered in your letter.

Introduction

Very clear and succinct introduction. Would you be able to elaborate slightly though on the pathway of mechanism by which probiotics affecting the GI tract would then impact skeletal muscle physiology and functionality. You could also elaborate on the reasoning for adding a heat treated arm of the study.

Thank you very much for your suggestions. We have added and rephrased the sentences (p.1-2, lines 39-50).

Methods

Line 61-62 - how was food intake monitored?

Thank you for your advice. We have added the text (p.2, lines 69-70). Food intake was monitored by the difference in weight between the food put into the cage and that remaining at the end of 24 hours.

Results

Table 1 - it is not clear what the P value represents? If all three conditions were compared to Control, for which does the P value represent? If any condition is significantly different to control, could you not indicate that in the relevant column?

We apologize for the confusion. The P value in Table 1 was removed.

Line 135 - 143 - could you provide individual p values where you describe differences to control group. You have described that plantaris and gastrocnemius relative weights increased, yet the table shows p > 0.05, suggesting no significant difference.

Thank you for your advice. We have rephrased (p.6, lines 147-149) to avoid the over-speculation.

Discussion

Line 221 - I think it would be more accurate to describe that Akt signalling was associated with increases in soleus mass.

The sentence was replaced (p.9, lines 229-230).

Line 222 - You did not assess mitochondrial biogenesis, so can only speculate that it would have been enhanced, along with mitochondrial energy production.

The sentences were rephrased (p. 9, lines 231-234).

224 - Again, while you can speculate that these adaptations are the cause of the increase in grip strength, you can only speculate on this association. However, I believe you need to make it clearer how increases in oxidative capacity would benefit muscular strength?

Thank you for your suggestion. Since some sections were likely to cause misunderstanding, the title and text were rephrased (p.9, lines 229-234) to better clarify our meaning. We also think that increases in oxidative capacity might not directly contribute to enhancement of grip strength.

Again, thank you for giving us the opportunity to strengthen our manuscript with your valuable comments and queries. We have worked hard to incorporate your feedback and hope that these revisions persuade you to accept our submission.

Sincerely,

Kazuya Toda.

Round 2

Reviewer 1 Report

While I appreciate the authors' consideration of many of my concerns, I have serious reservations regarding their analytical technique. I do not believe Dunnett's multiple comparison test (a test of all versus control) to be appropriate, particularly if the authors' aim to make claims such as (page 6; lines 145-146) "A significant increase in weight of the soleus was observed in the B-3HK group." Citing the use of Dunnett's test in previous papers is not an excuse for lack of scientific rigor.

Moreover, the authors continue to perpetuate the claim that (page 8; lines 204-205) "B-3HK significantly increased grip strength". Stating that B-3HK "increased" implies a change, and this conclusion cannot be drawn without a baseline measure.

Based on these serious statistical/methodological flaws I simply cannot recommend the paper for publication.